# Emotional Intelligence and the Different Manifestations of Bullying in Children

**DOI:** 10.3390/ijerph17238842

**Published:** 2020-11-28

**Authors:** Jesús M. Alvarado, Amelia Jiménez-Blanco, Teresa Artola, Santiago Sastre, Carolina M. Azañedo

**Affiliations:** 1School of Psychology, Complutense University of Madrid, 28223 Madrid, Spain; ajimblangr@ucm.es; 2Department of Psychology, Villanueva University, 28034 Madrid, Spain; tartola@villanueva.edu (T.A.); ssastre@villanueva.edu (S.S.); cmartin@villanueva.edu (C.M.A.)

**Keywords:** ability emotional intelligence, cinema scenes, socio-emotional development, bullying, emotional knowledge

## Abstract

The main objective of this research was to help clarify the relationship between ability emotional intelligence (AEI) and bullying in children. Bullying is a maladaptive behaviour that generates severe adverse consequences in the school environment and is a matter of growing concern in the educational community. To investigate the relationship between AEI and bullying, we administered two tests to a sample of 329 students (52.9% girls) aged between 8 and 12 years old (*M_age_* = 9.3; *SD* = 1.2). AEI was assessed using a test based on the interpretation of cinema scenes (EMOCINE). EMOCINE was designed to measure two of the primary factors considered in Mayer and Salovey’s ability model: emotional perception and emotional understanding. Furthermore, we administered a measure of bullying and school violence (AVE), which provides a global index of bullying, as well as a measure of its intensity, by considering eight scales or types of victimisation (harassment, intimidation, coercion, threats, social blocking, social exclusion, manipulation and aggression). The results show that age had a statistically significant effect on measures of bullying, while gender showed an interaction with victimisation types. A reduction in bullying behaviours was observed as the age of children increased, while gender-based analyses revealed different patterns in bullying behaviours. Regarding EI, it was observed that students with high AEI scores presented the lowest levels in both global bullying indexes and the victimisation types. Consequently, AEI seems to have important implications for bullying behaviours, and therefore, interventions aimed at the evaluation, training and development of AEI might offer the educational community the possibility of preventing or redirecting bullying situations.

## 1. Introduction 

The main goal of this research was to help clarify the relationship between emotional intelligence (EI) and bullying in children, conceptualising and measuring EI as an ability (AEI) [1] or a part of intelligence rather than as a personality trait. We assessed school bullying from the victim’s perspective, analysing the frequency with which some schoolchildren are subjected to psychological bullying [2]. We also assessed the relationship between victimisation and the variables of age and gender. In line with these objectives, below we give a brief review of the conceptualisation of EI, its measurement in the child population and studies that have related EI and bullying. 

### 1.1. EI and Its Measurement in the Child Population 

There is debate surrounding the conceptualisation of EI, with some considering it a part of intelligence (AEI) and others who defined it as a personality trait (trait EI). Thus, the broad spectrum of instruments developed [3,4,5], and consequent disparate and sometimes contradictory results are partly explained by the conceptual disparity surrounding the construct of EI.

The conceptualisation of trait EI as an element or facet of personality situates the construct in the emotional part of the personality [6,7,8,9,10]. In any event, self-reports used from this perspective, reveal individuals’ beliefs about whether they can perceive, discriminate and control their emotions. This indicator is called the perceived or self-report EI index (compilation of questionnaires or tests from this perspective in [11]).

Meanwhile, the AEI perspective relates the emotions to cognitive processes and abilities [12,13,14,15,16,17,18], contending that most modern theories of emotion view cognitive processes as an essential element of emotion. Thus, cognition or reason forms an integral part of how emotions are generated, perceived, understood, analysed and regulated. Salovey and Mayer [19] were the first to conceptualise EI as an ability, and subsequently, in 1997, they presented a model based on the emotional processing of information. 

The classic conceptualisation of intelligence implies that its measurement should meet a series of criteria or requirements: according to the so-called correlation criterion, for EI to be considered as such, the results of its measurement must correlate positively with the results of other intelligence tests [13]. In a meta-analysis of the relationship between EI and other types of intelligence in the university and non-university population, Kong [20] reported the correlations between the MSCEIT [14], the most frequently used measure of AEI, and different types of intelligence: the MSCEIT presented a correlation of 0.40 with the Weschler test on its verbal scale; of 0.28 with the verbal SAT and 0.30 with the Raven test, a non-verbal instrument that measures fluid intelligence. 

Whatever its facet, intelligence is linked to the criterion of development [13]. This criterion is defined as the development of intelligence throughout life. Cabello, Navarro, Latorre and Fernández-Berrocal [21] suggests that emotional management improves with age. Older adults make greater use of anger regulation and re-evaluation strategies but less use of ineffective regulation techniques such as suppression [22,23]. Nevertheless, older adults experience deficits in emotion recognition tasks, albeit not in understanding verbal material with emotional content [24]. The empirical results seem to indicate that not all the primary factors in Mayer and Salovey’s model [12] develop throughout life in the same way.

Moreover, as León Rodríguez and Sierra Mejía [25] have observed, emotional understanding and regulation occur at an early age. Therefore, the criterion of development that characterises any type of intelligence in order for it to be conceptualised as such is best verified in samples of the child population rather than later age ranges, in which intelligence measurement scores tend to stabilise, as occurs in the case of classical intelligence and its corresponding IQ. Studies have shown that EI increases progressively with age [26,27,28]. 

This conceptualisation of AEI was operationalised [13] with the design of the MSCEIT ability- or performance-based test (adult version [14,17] and youth version [15,29]). The test divides EI into four primary hierarchical factors corresponding to the four branches indicated in the definition of the construct: emotional perception, facilitation, understanding and regulation [12]. 

AEI tests in Spanish aimed at a child population include the interesting Botín Foundation Emotional Intelligence Test for Children (TIEFBI), based on a test by the same authors aimed at adolescents (TIEFBA, [30]). The TIEFBI test, which is not yet available, is aimed at children aged between 30 months and 12 years old and includes three of the primary factors in Mayer and Salovey’s [12] theoretical model: emotional perception, understanding and regulation, excluding emotional facilitation [3]. Until recently, this test was the only instrument in Spanish for measuring ability—rather than trait—EI. In 2019, however, the EMOCINE test (EMOtion in CINEma scenes [1]) was developed to measure AEI in the child population. It is based on viewing fifteen film scenes with high emotional content aimed at a child audience, and the present study was one of the first to administer this tool.

### 1.2. EI and Victimisation in Bullying

School bullying has been defined as a type of violence in which one or more students intentionally and repeatedly harm other students who are physically or psychologically weaker, i.e., there is an unequal power relationship [31]. For a long time, this violence is expressed through aggressive behaviour, which reflects an imbalance of power between bully and victim, and aggression becomes abuse [32,33,34]. Students assume different roles in bullying: some become bullies who engage in aggressive behaviour, others become victims of repeated abuse, while still others inhabit both roles throughout their school lives.

In a systematic review and meta-analysis conducted in 2019, Zych, Ttofi and Farrington [31] identified 53 studies that included the keywords “callous-unemotional”, “empathy”, “bullying”, “aggression”, “aggressive”, “victimisation”, “emotion”, “emotional intelligence” and “emotional awareness”. This comprehensive review concluded that two interrelated and very promising approaches to bullying had been proposed. One of these focuses on the relationship between bullying and empathy, while the other explores the relationship between bullying and trait EI. The main objectives of the present study are related to the second of the above options but considering EI as intelligence or ability. Studies on the association between trait EI and bullying have proved a very promising avenue of enquiry, but one which has received limited research attention [31]. As expected, trait EI is associated with bullying. As an initial theoretical framework for our study, the scientific evidence indicates a significant negative correlation between trait EI and victimisation by bullying [35,36,37,38,39]. This negative correlation between EI and victimisation suggests that victims lack the personality traits that would enable them to solve problems, whereas children with high EI seem to have the tools necessary to respond when they are bullied [40] effectively. 

Our prior review of the literature justified the inclusion of the relationship between victimisation and next variables: age and gender. At present, the consensus seems to have been reached regarding the variable of age, which, as with trait EI shows a negative correlation with bullying and victimisation, with a reduction in the latter as subjects age increases [32,41,42,43,44,45,46,47,48]. Results concerning the influence of gender on victimisation, however, are more controversial. Some studies have found no gender-related differences [46], but most have reported that this variable significantly affects victimisation in school [32,43,45]. 

To date, the absence of AEI measures for the child population has hindered study of the relationship between bullying and cognitive development at each developmental stage of AEI. In Mayer and Salovey’s [13] hierarchical model, perception and facilitation are the basic “branches” on which emotional understanding and regulation are constructed. EMOCINE was designed to differentiate between subjects with low levels of AEI and descriptive-type responses (i.e., subjects in stages of perceptual development) and those with high AEI in a stage of emotional understanding, which cannot be achieved without prior perception or identification of the emotions that form the basis of an emotional situation (see the AEI developmental model for EMOCINE, [49]). Since bullying situations can be seen as situations of emotional conflict, subjects whose AEI has developed to the level of understanding would be expected to have the ability to deal more efficiently with bullying situations. In terms of gender, girls’ higher AEI scores and more rapid development may be related to the pattern observed by Olweus [32,45] as regards girls’ lower experience of victimisation and steeper reduction in victimisation with age.

### 1.3. Objectives and Hypothesis

The present study aimed to elucidate the relationship between AEI and bullying from the perspective of the role of the victim, not the bully. In this relationship, our theoretical position was based on the conceptualisation of the emotional construct as intelligence or ability, according to the theoretical model of Mayer and Salovey [12]. If cognitive development enables victims to avoid or control possible situations of bullying, then rates of bullying should decrease with age as AEI increases (hypothesis 1). The developmental stage of AEI positively affects both the degree and the facets or dimensions of bullying perceived by the victims (hypothesis 2). We also sought to investigate the influence of the variable of gender, both on general scores and on the different scales and dimensions of victimisation (hypothesis 3). 

## 2. Method

### 2.1. Participants

We used a convenience sample of 329 students (52.9% girls) with a mean age of 9.3 years (SD = 1.2) who were attending a publicly-funded state school located in a suburb of the city of Madrid (Spain). Participants were in their 3rd to 6th year of primary education and were aged between 8 and 12 years old. 

### 2.2. Procedure

Students collectively completed EMOCINE and AVE. The School of Psychology Ethics Committee at the Complutense University of Madrid approved this study (Ref. 2018/19-013). One month before administering the tests, students’ parents received a letter signed by the head teacher describing the research project and enclosing an informed consent form for them to sign. To ensure participant anonymity and confidentiality, each student was assigned a code that was written on each of the tests. The information was recorded and stored in a safe place in such a way that researchers would be unable to identify each student.

### 2.3. Instruments 

EMOCINE (EMOtion in CINEma scenes, a measure of AEI for the child population) [1] is based on viewing fifteen film scenes with high emotional content, aimed at a child audience. Participants are given a response sheet and asked to pay close attention to the video they are going to watch. The instructions are explained at the beginning of the video, but in addition, the researchers administering the test ensure that all subjects have understood these instructions before starting the first scene. Instructions and response options are pre-recorded to avoid any possible verbal contamination that might be caused by children reading them. First, subjects listen to the instructions, and then the selected scenes are viewed one by one. After each scene, the individual must select one of the three pre-recorded response options. The three response options are based on categories identified in a pilot study, in which three types of answers were detected: the correct one, where individuals show a developmental stage of AEI that enables them to perceive the emotional elements and interpret them correctly in all their complexity; a second response reflecting a stage in which the individual perceives and interprets the emotional elements but is not yet able to understand them in all their complexity; and a third and final response corresponding to an essential emotional perception without any capacity for interpretation. The two dimensions that EMOCINE measures correspond to the emotional perception and understanding included in Mayer and Salovey’s model [12]. The test lasts around forty minutes, and none of the scenes lasts more than two minutes. EMOCINE yields a global AEI score and also classifies the subjects into three levels of AEI development (low AEI or naive subjects, intermediate AEI or developing subjects and high AEI or sensitive subjects). The test shows a one-dimensional structure with marginal reliability of 0.74 [49].

AVE (bullying and school violence [2]). This test consists of two parts: one measures bullying behaviour, and the other assesses psychological or psychosomatic symptoms generated by the harm that generally accompanies bullying. In this study, we administered the first part of the AVE corresponding to the assessment of bullying, which consists of eight scales and two global dimensions. The global bullying index and the intensity of bullying comprise l00 items divided into eight scales to measure different types of victimisation: harassment, intimidation, coercion, threats, social blocking, social exclusion, social manipulation and aggression. The AVE presents satisfactory internal consistency reliability, which ranges from 0.78 to 0.95 [2].

## 3. Results

### 3.1. Age and Gender 

A multivariate analysis of variance (MANOVA) was performed for the dependent variables of global bullying index (GBI), global bullying intensity index (GBII) and AEI scores. Given the high correlation between the dependent bullying variables, assessed from the victim’s perspective, GBI and GBII (*r* = 0.85, *p* < 0.001), we used Roy’s Largest Root as a contrast statistic and observed a significant main effect of age (*F*(3, 321) = 21.094, *p* < 0.001, η_p_^2^ = 0.165). The variable of gender did not reach statistical significance although it did show a small effect size (*F*(3, 319) = 1.862, *p* = 0.136, η_p_^2^ = 0.017). Meanwhile, the gender x age interaction was statistically significant (*F*(3, 321) = 3.998, *p* = 0.008, η_p_^2^ = 0.036). Additionally, univariate analyses show only a significant effect for age: GBI *F*(3, 321) = 12.965, *p* < 0.001, η_p_^2^ = 0.108, GBII *F*(3, 321) = 4.039, *p* = 0.008, η_p_^2^ = 0.036, AEI *F*(3, 321) = 6.885, *p* < 0.001, η_p_^2^ = 0.060. The reason why gender x age interaction is obtained in the multivariate analysis and not in the univariate analyses is probably the statistical power. In the multivariate analysis (combined variable) the statistical power of 0.835 was adequate, while in the univariate analyses it was below 0.30 in all three cases due to the limited sample size for these analyses, although certain trends can be observed (see Figure 1).

Figure 1 shows that the GBI and GBII bullying indexes for victimisation declined as age increased, in contrast to the scores for AEI, which increased with age. In terms of the age x gender interaction, it can be seen that the pattern of a decline with age for GBI was greater in girls than in boys, and in turn, the AEI scores showed a greater increase with age in girls than in boys. With regard to the intensity of victimisation measured by the GBII index, except for the youngest participants, where girls reported a higher intensity, both groups reported a similar level of decrease in bullying intensity as age increased.

### 3.2. Type of Victimisation and AEI

In a second analysis, we assessed the relationship between AEI and types of victimisation, taking the three levels of AEI identified by EMOCINE as the independent variable. 

We performed a repeated measures analysis of variance (ANOVA) for the variable ‘type of victimisation’ to assess the possible effect of the variables AEI, age, and gender, including age as a covariate. 

The analysis revealed a significant main effect of type of victimisation (*F*(7, 2198) = 18.390, *p* < 0.001, η_p_^2^ = 0.055), of AEI (*F*(2,314) = 4.894, *p* = 0.008, η_p_^2^ = 0.030) and of the covariate age (*F*(1, 314) = 28.790, *p* < 0.001, η_p_^2^ = 0.084). Gender was not found to have a significant effect (*F*(2, 314) = 1.921, *p* = 0.148, η_p_^2^ = 0.012), although there was a significant interaction between type of victimisation and gender (*F*(2, 314) = 2.247, *p* = 0.028, η_p_^2^ = 0.007). 

Figure 2 shows that harassment, aggression, social manipulation and social blocking were the four most frequent types of victimisation. 

Overall, boys presented slightly higher levels of victimisation than girls (*M_girls_* = 1.65 vs. *M_boys_* = 1.75), but this was not statistically significant as there was a compensatory effect resulting from the interaction shown in Figure 3, whereby harassment, aggression and intimidation were more frequently reported by boys, whereas social blocking and social exclusion were more frequent in girls (see Figure 3). In any event, there is considerable overlap of the error bars, which indicates a weak interaction. 

AEI showed a main effect on victimisation, whereby subjects with high AEI (*M* = 1.22, *SE* = 0.19) presented fewer statistically significant victimisation behaviours than those with intermediate AEI (*M* = 1.89, *SE* = 0.15) and low AEI (*M* = 1.99, *SE* = 0.21). A post-hoc analysis using the Bonferroni test revealed that the differences in victimisation between the groups with low and intermediate AEI were not statistically significant, but they were significant between the high and intermediate AEI groups (*p* = 0.018) and the high and low AEI groups (*p* = 0.021). The interaction of victimisation and AEI was also statistically significant (*F*(14, 2198) = 2.606, *p* = 0.001, η_p_^2^ = 0.016) (see Figure 4).

Figure 4 shows a downward trend for the eight victimisation behaviours as AEI increased; this was more pronounced for aggression, social manipulation and social blocking, the three behaviours against which AEI exerted a more significant protective effect. 

## 4. Discussion

The main goal of this study was to demonstrate the relationship between emotional perception and understanding and victimisation in children, based on the premise that both constructs are related to cognitive development and consequently to age, and also to explore possible differences in victimisation according to gender. 

The relationship between age and victimisation (hypothesis 1) was direct and negative in our study, corroborating the results of previous studies [32,41,42,43,44,45,46,47,48]. This result suggests that the construct of victimisation has a negative relationship with age since victimisation showed a statistically significant decline as age increased. The reasons for this inverse relationship between age and victimisation are, of course, multiple and complex. However, there is consensus in the literature regarding the significant negative relationship between age and victimisation, the explanation for the decrease in the latter as the former increases remain unresolved. 

EI as an ability rather than a personality trait may explain some of this relationship, as there is empirical evidence for the progressive increase in AEI as individuals age [21,26,27,28,49]. Higher trait EI scores are associated with lower levels of victimisation and vice versa [35,36,37,38,39]. The present study shows for the first time that AEI in the child population has a significant and negative relationship with victimisation (hypothesis 2). Further research is required to relate EI and victimisation from an AEI perspective using instruments that measure performance or achievement. Suppose AEI is considered one more form of intelligence, thus meeting the criterion of the development just as other types of intelligence do. In that case, this would partially explain the relationship between age and victimisation, since AEI would predict the decline in victimisation as age increases. Other elements or factors would also be involved in this negative relationship between age and victimisation. Hence, further research is required to investigate these elements or factors, including longitudinal studies, to establish causal relationships that would enable us to implement school bullying and victimisation prevention programmes based on scientific evidence.

Studies that have considered the influence of gender on victimisation have been more controversial (hypothesis 2). Although some research has found no evidence that this variable exerts an influence on school bullying or victimisation [46], most studies have reported that gender significantly affects victimisation [32,43,45,50]. Boys report a higher percentage of victimisation than girls [45,50]. The most notable differences occur in the upper school years, as there is a significant decline in victimisation of girls from the ages of 11 or 12, whereas for boys, it stabilises between the ages of 12 and 14 [32]. Our study, with participants aged between 8 and 11, showed similar indices by gender up to age 9, when more victimisation occurred in boys than in girls. Age implied a decline in victimisation after the age of 9, which occurred much more gradually in boys than girls. The protective effect of AEI against victimisation becomes more marked in girls from the ages of 8 or 9, as differences in AEI scores widen with age and boys present lower levels. Thus, a higher AEI in girls may partly explain the more significant decline in victimisation scores than occurs in boys as age increases.

The smaller decrease in victimisation scores for boys may be related to the type of response that victims adopt [43] since boys tend to respond to provocation with reactive aggression, whereas girls do so proactively. Reactive aggression is a defensive response to provocation and is accompanied by anger, whereas a response based on proactive aggression, usually adopted by girls, is a targeted action, deliberately formulated and unaccompanied by anger. As noted, these differences have been explained in terms of the type of response according to gender: reactive aggression in the case of boys and proactive aggression in the case of girls. Type of response could be linked to the degree of cognitive development in boys and girls according to the variable of age. With earlier cognitive development, girls’ response to bullying would enable them to avoid or redirect it more successfully than boys, who at the same age show less cognitive development. Furthermore, as indicated earlier, the higher AEI scores for girls than boys [28,29,49,51] might partially explain the relationship between age and victimisation. 

Although our study did not reveal a significant main effect of gender, we did find that gender differentially affected some behaviours. Harassment, aggression and intimidation were more frequent in boys, whereas girls experienced a higher incidence of social blocking and exclusion. These gender-related differences in the type of victimisation require further research to confirm and explain our results. Beyond gender-related differences in overall victimisation rates, future research should also investigate the influence of AEI on the different facets of victimisation and the associated gender-related differences. 

## 5. Limitations

As study limitations, it should be noted that our results are restricted to the instruments used to measure AEI and bullying. It would be very interesting to have measures of AEI facilitation and regulation in order to determine which components of AEI are most predictive of victimisation and bullying behaviour. In addition, longitudinal studies are required to obtain a more accurate assessment of the AEI developmental model and its relationship to bullying. Lastly, before the results can be generalised, it will be necessary to replicate this study using more representative probability samples in order to assess the effect of different socio-demographic variables or the socio-economic context of the school on type and intensity of bullying. 

## 6. Conclusions

AEI can exert a protective effect against bullying; this result could be due to a higher ability to understand and therefore to manage or regulate these situations of emotional conflict more efficiently. This finding has important practical implications as it paves the way for implementing activities or programmes aimed at developing AEI in order to reduce the problem of bullying by helping children who experience it to improve emotional control and regulation in these situations.

## Figures and Tables

**Figure 1 ijerph-17-08842-f001:**
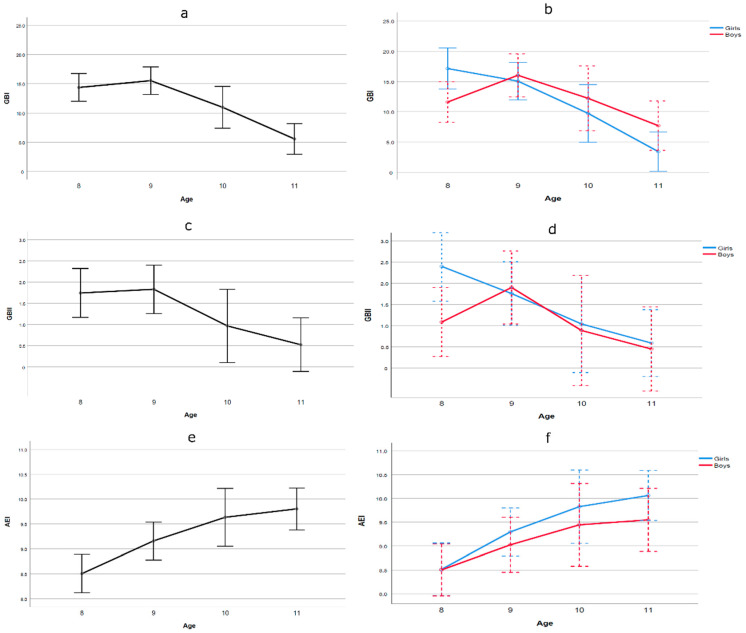
Age and age by gender on global bullying index (GBI, figures **a** and **b**), global bullying intensity index (GBII, figures **c** and **d**) and ability emotional intelligence (AEI, figures **e** and **f**). Error bars 95% CI.

**Figure 2 ijerph-17-08842-f002:**
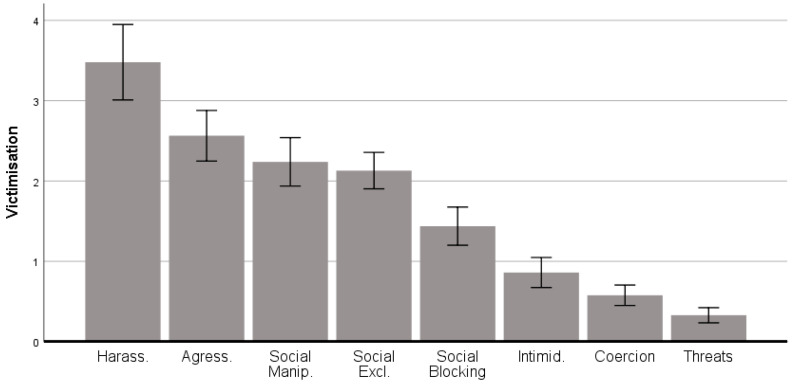
Types of victimization. Error bars 95% CI.

**Figure 3 ijerph-17-08842-f003:**
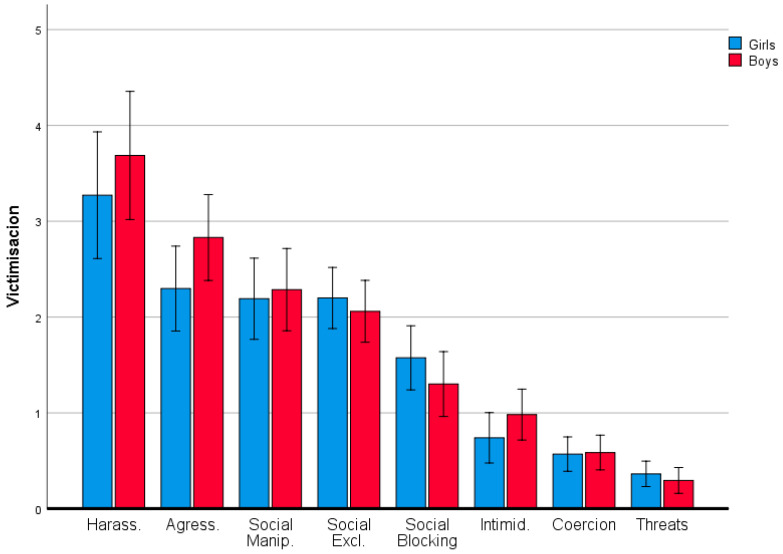
Types of victimisation by gender. Error bars 95% CI.

**Figure 4 ijerph-17-08842-f004:**
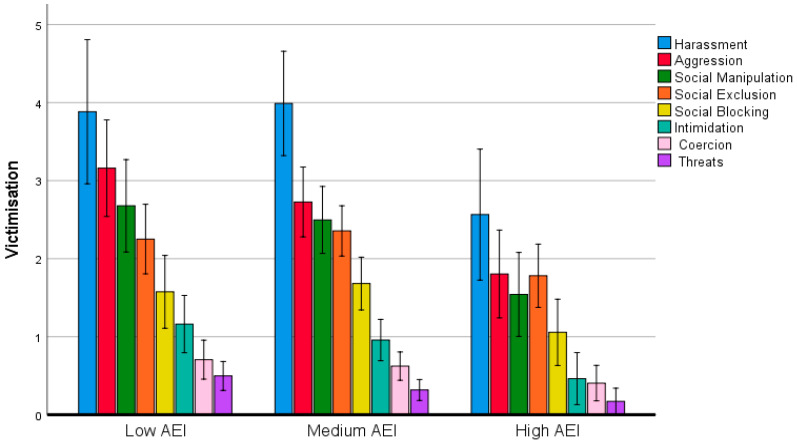
Types of victimisation by AEI group. Error bars: 95% CI.

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
