# Peer review of "Emotional Intelligence and the Different Manifestations of Bullying in Children"

_ijerph, 2020, doi:10.3390/ijerph17238842_

Round 1
Reviewer 1 Report
Overall the study is interesting and from an important topic. There were some typos (IE in the Abstract).
Some improvement could be made. For instance, it should be clearly described in the titles of the tables that the results involve bullying behavior by the subjects themselves, not that the subjects are victims of bullying. The results should be clarified in this sense as not all of the readers will be able to read the text in a deep manner.
It should be mentioned if EMOCINE is a widely used and accepted method. Was this among the first studies to utilise it?
Author Response
Thank you very much for your review. In light of your comments, we have made the following changes:
The abstract has been corrected by replacing “IE” with “EI”.
We have added further clarification in the figures indicating that the bullying indices refer to victimisation.
EMOCINE is a promising tool developed in 2019, and the present study is one of the first to administer it, as indicated in lines 93-96.
Reviewer 2 Report
The results of this study highlight three mains findings: (1) age had a statistically significant effect on bullying, while sex showed an interaction with victimisation types; (2) a reduction in bullying behaviours was observed like the age of children increased, while sex-based analyses revealed different patterns in bullying behaviours; (3) students with high EI scores presented the lowest levels in both global bullying indexes and the 28 victimisation types (the most relevant result from my point of view). A total of 329 Spanish primary students participated in this study.
Although the topic, the developmental stage of the participants, and the use of the EMOCINE instrument are strong points of this work, the whole manuscript present major weaknesses that I explain below.
INTRODUCTION
The introduction comprises two main parts: conceptualization of EI and its measure with child population and the relationship between emotional intelligence as a personality trait and victimisation in bullying
In general, the conceptualization of EI section is too long and it could shorten. Also, more information could be added to better explain the association between age, gender, and EI with bullying.
L. 137-138: I consider EMOCINE a good tool to evaluate EI in children under 9-10 years old. It is a better option than a self-report questionnaire. However, the authors state “The present study aims to help remedy this gap by administering the EMOCINE test” (L. 137-38) with respect to different consideration of the concept EI at early developmental stages and other issues. I think the authors should better explain why this instrument covers this gap. This is not clear to me neither for the readers.
L-143-44: The authors write: “Some studies have found no sex-related differences 143 [46], but most have reported that this variable significantly affects victimization in school [32, 43, 45].”. In which direction does sex affect to victimization in these studies?
L. 146-147: “The present study aimed to elucidate the relationship between EI and bullying from the 146 perspective of the role of the victim, not the bully” Is the literature review of studies in the introduction considering this view? The authors should be more specifically focused on the victim perspective.
HYPOTHESIS:
L. 149-151: Hypothesis 1 writing is unclear. It is necessary to rewritten “Age is a significant variable concerning rates of bullying, since the latter decreases as ability EI increases, enabling victims to avoid or control possible situations of bullying (hypothesis 1).”
METHODS
How was the sample selected? More descriptive information. Where are the schools located? What kind of sampling was used?
The explanation of the procedure is scarce. How was EMOCINE administered? What were the participants told? How were anonymity and confidentiality assured? When were the informed consent forms signed? How long did the whole procedure take?, etc.
RESULTS
There is a lack of important statistical analyses. For example, the interaction sex*age is significant, but not post-hoc analyses were run to know what differences between groups are significant in each GBI indexes.
L. 219-223: “Figure 2 shows that harassment, aggression, social manipulation and social blocking were the four most frequent types of victimisation. Although sex did not have a significant main effect, we did find that it differentially affected some behaviours, whereby harassment, aggression and intimidation were more frequently reported by boys, whereas social blocking and social exclusion were more frequent in girls (see Figure 3).”
If there is not a significant effect of sex, how do the authors can state that “harassment, aggression and intimidation were more frequently reported by boys, whereas social blocking and social exclusion were more frequent in girls”. I am sorry but this is utterly incorrect. This can only be stated if there are statistically significant differences between the groups.
L. 209-209: “…we assessed the relationship between EI and types of victimisation, taking 208 the three levels of EI identified by EMOCINE as the independent variable.” How were these groups formed? The authors could have kept the variable as quantitative (which restrains less the variance) and have used a correlational approach (a regression model with controls). Why did the authors use this method and what did they exactly do?
DISCUSSION
Although I can see some future research lines, the manuscript does not describe any limitation of this work (and there are some clear ones) nor any practical implication.
FINAL COMMENT
I would like to encourage the authors to keep working on this important research area. I hope these comments help the authors to improve the quality of their work.
Author Response
INTRODUCTION
Thank you very much for your comments and suggestions, which have been very helpful and which we have incorporated as indicated below:
We have condensed the conceptualisation of EI and substantiated the relationship between EI ability and victimisation, as well as the effect of age and gender (lines 120-139).
Question about L. 137-138. The wording has been improved and clarified to explain that the problem encountered when measuring EI ability in children is the lack of EI-ability instruments specifically adapted for this age group, and that EMOCINE was developed to overcome this shortcoming (lines 87-96).
Question about L. 143-44. It is true that we did not report the direction of the effect of gender, but we have now indicated that victimisation is greater in boys than in girls, and that the decline in victimisation with age occurs more rapidly in girls (lines 137-139).
Question about L. 146-147. As mentioned in the text and evidenced in the systematic review and meta-analysis by Zych, Ttofi and Farrington (2019), very few studies have linked EI and victimisation (lines 105-111). Nevertheless, we conducted an extensive literature search to identify and cite those that have been carried out.
Question about L.149-151 (hypotheses). In line with the reviewer’s recommendation, the wording of the hypotheses has been changed (lines 144-149).
METHODS
The study was conducted using a non-probabilistic convenience sample, and this has now been included among the study limitations. We have indicated the location of the school and have expanded our description of the procedure by providing information about the application, anonymity and confidentiality (new text in lines 155-173).
RESULTS
Since post-hoc analyses are only available for the main effects, the interactions identified by the ANOVA are visualised by means of graphs (graphical method). However, to facilitate assessment of the significance of the differences between levels using the graphical method, the 95% CI error bars are shown.
Comment L. 219-223. We thank the reviewer for the suggestion to conduct post-hoc analyses to assess main effects (see lines 244-247), as these illustrate that the only difference between the three levels of EI with respect to victimisation occurs with the high level of EI, which is highly relevant from a theoretical perspective, as explained in the text. Regarding the interaction between type of victimisation and gender identified in the ANOVA, we have examined the error bars and agree with the reviewer that this is not very significant. Consequently, we have changed the wording to indicate that this result requires further research and evidence.
Comment L. 209-209. EMOCINE identifies the number of correct responses that show emotional sensitivity and also enables subjects to be classified according to three levels of emotional development (low, intermediate and high) as explained in the section on instruments. This classification follows a developmental model of EI (Sastre et al., 2019; Jiménez-Blanco et al., 2020) and has been shown to be a more explanatory measure of the construct. Although —as suggested by the reviewer— EI could have been included as a covariate or its effect could have been assessed using a regression model, it was important for the purposes of our article to assess the level of victimisation for each developmental level of EI (low, intermediate and high). As the analyses reveal, victimisation decreased significantly only at high levels of EI.
DISCUSSION
We have included a section on study limitations (lines 311-319) and conclusions (lines 321-327).
We reiterate our thanks, and believe that the reviewer’s comments have significantly helped to improve our article.
Reviewer 3 Report
Introduction
- My main concern is about the introduction: Authors extend too much the explanations of the different models of EI, while it is not the main focus of the study. The introduction would benefit of the deepening on the rationale of relating EI and bullying (and their subscales) and the different variables employed such as age and gender.
- Which trait EI subscales has been preciously related with bullying?
- There are no studies of ability emotional intelligence and bullying, but what about aggression? Some ideas could be included as in the discussion this is addressed.
- Hypothesis should indicate the expected results. In addition, results and discussion should follow the same order than the hypothesis in order to make the paper consistent and more easily understood.
- Page 2, line 55: “tests aimed at measuring skills or competencies —the so-called performance tests— _would be measuring trait EI and not a part of intelligence” --> Performance tests are devoted to measured EI as a type of intelligence.
Instruments:
Does the EMOCINE yields an EI score for each of the two branches measures? In that case, why it has not been analyzed?
Results
Author should briefly explain the statistical results found in each section.
Discussion
- The discussion would benefit from an initial paragraph summarizing the aim of the study.
- Page 7, line 245: “The present study shows for the first time that ability and not trait EI in the child population also has a significant and negative relationship with victimization”--> with this sentences it seems that you have employed both, the ability and the trait EI instrument, and you only found a significant relationship with the first. In addition, I do not really understand what you mean in “At a theoretical level, further research is required to relate EI and victimisation from an ability EI perspective using instruments that measure performance or achievement, in line with this theoretical position”.
- Limitations should be included: e.g., the EI instrument hasn’t got some important branches of EI, for instance, the managing branch. How could each branch affect the results?
- Future lines should be included: which are the clinical implications? There already exist some EI interventions (e.g., INTEMO program) that has shown to reduce aggression in adolescents.
- The article would benefit from a final conclusion.
References
- Some references should be revised.
Others comments:
- Some EI terms have been changed by IE.
- I am not an English native speaker, but I think English should be revised.
Author Response
Thank you very much for your review. In light of your comments, we have made the following changes:
INTRODUCTION
We have condensed the conceptualisation of EI and substantiated the relationship between EI ability and victimisation, as well as the effect of age and gender (lines 120-139).
Although it would be very interesting to investigate the subscales of EI trait with harassment, this question is beyond the scope of the present study on EI ability; nevertheless, we will consider it in future research.
To expand the state of the question regarding harassment and its relationship with EI, we have cited the recent systematic review by Zych, Ttofi and Farrington (2019) (lines 105-111).
We have revised the wording of our hypotheses so that each of them now reflects the expected results (lines 141-151).
(Page 2, line 55) The paragraph indicated was potentially confusing, as it referred to the opinion of authors who defend the EI trait perspective. Consequently, in line with the reviewer’s suggestion to shorten the section on the conceptualisation of EI, it has been removed.
Instruments: EMOCINE was designed to differentiate between subjects with low levels of EI and descriptive-type responses (i.e. subjects in stages of perceptual development) and those with high EI in a stage of emotional understanding, which cannot be achieved without a previous perception or identification of the emotions that are the basis of an emotional situation (lines 127-134).
RESULTS
We thank the reviewer for this observation, and we have added a brief explanation of each result obtained.
DISCUSSION
We have added an initial paragraph summarising the study objectives (Lines 255-258).
Page 7, L. 245. We agree that references to “EI trait” and “theoretical level” were indeed confusing and these have therefore been eliminated.
As suggested by the reviewer, we have added a “study limitations” section before the “conclusions” section (Lines 311-319)
In addition, we have added a “conclusions” section indicating some of the practical implications of our findings, such as intervention to improve EI in order to reduce victimisation (Lines 321-227)
REFERENCES AND OTHER COMMENTS
We have revised the references and the text has also been carefully revised to eliminate all possible typographical, spelling or grammatical errors.
Round 2
Reviewer 2 Report
L. 198-200: "A multivariate analysis of variance (MANOVA) was performed for the dependent variables of 198 global bullying index (GBI), global bullying intensity index (GBII) and EI ability scores." Since the authors used a MANOVA, the authors have to run follow-up ANOVAs to make sure which differences correspond to which dependent variable. This is an omnibus test, so it is necessary to do follow-up analyses.
In addition, in L. 204-205 it is said taht "the gender x age interaction was statistically significant (F(3,321) = 204 3.998, p=0.008, η2partial = 0.036)". This means that at least one of the gender x age interaction in one of the dependent variables (global bullying index (GBI), global bullying intensity index (GBII) and EI ability scores) would be significant. In this case, post-hoc test with the Bonferroni adjustment should be run too.
L. 221-222: "We performed a repeated measures analysis of variance (ANOVA) for the variable ‘type of victimisation". From my understanding, the authors have used a MANOVA, although the have used the interface (SPSS most likely) for a repeated measures analyses, but conceptually they have run a MANOVA not a repeated measures ANOVA (there is no time 1, time 2, ...).
After including this important corrections, the manuscript can be accepted for publication.
Author Response
Regarding the first two questions, ANOVA by variable has been included as requested, including the following paragraph (L 199-205):
Additionally, univariate analyses show only a significant effect for age: GBI F(3,321) = 12.965, p <0.001, η2partial = 0.108, GBII F(3,321) = 4.039, p = 0.008, η2partial = 0.036, EI ability F(3,321) = 6.885, p <0.001, η2partial = 0.060. The reason why gender x age interaction is obtained in the multivariate analysis and not in the univariate analyses is probably the statistical power. In the multivariate analysis (combined variable) the statistical power of 0.835 was adequate, while in the univariate analyses it was below 0.30 in all three cases due to the limited sample size for these analyses, although certain trends can be observed (see Figure 1).
Regarding the third question, the reviewer indicates one of the main uses of ANOVA for repeated measures (i.e. study any change in subjects from one time point to another), however, this technique is also used for the comparison of related measures (i.e compare the effects of different treatments/conditions). ANOVA for repeated measures is the equivalent to the one-way ANOVA, but for related, not independent, groups (It is also an extension of the dependent t-test).
Thank you very much for your comments and suggestions
Reviewer 3 Report
- "Emotional intelligence ability" should be "ability emotional intelligence" and "EI ability", "ability EI".
- In my opinion, some information is not necessary: How the MSCEIT should be developed is not relevant nor the database of the systematic review.
- Line 149-151: "Main result was that ability EI exerts a protective 149 effect against bullying, explained by the more remarkable cognitive development of individuals with 150 greater EI" --> Should be expressed as an hypothesis.
- In the conclusion, as there is no causality in the study, the following sentence should be expressed in terms of correlations: "Children with high EI suffer less from bullying".
- Some references are not yet correct. For instance, the authors of "Pacheco, N. E.; Berrocal, P. F. Una guía práctica de los instrumentos actuales de evaluación de la inteligencia emocional" are not correctly referenced. There are more examples along this section that should be revised.
Author Response
The afore mentioned typo has been corrected in the abstract. Information on MSCEIT has been reduced (see lines 80-83). The mention of databases has also been removed.
About Lines 149-151. This sentence is part of the conclusions so it has been removed from the hypotheses.
The conclusion has been reformulated avoiding establishing causal relationships.
Errors in references have been corrected.
Thank you very much for your comments and suggestions